# Impact of Increasing Lower Body Negative Pressure and Its Abrupt Release on Left Ventricular Hemodynamics in Anesthetized Pigs

**DOI:** 10.3390/jcm11195858

**Published:** 2022-10-03

**Authors:** Birgit Zirngast, Leonhard Berboth, Martin Manninger, Helmut Hinghofer-Szalkay, Daniel Scherr, Lonnie G. Petersen, Nandu Goswami, Alessio Alogna, Heinrich Maechler

**Affiliations:** 1Department of Cardiac Surgery, Medical University Graz, 8036 Graz, Austria; 2Department of Cardiology, Charité—Campus Virchow-Klinikum, 13353 Berlin, Germany; 3Department of Internal Medicine, Division of Cardiology, Medical University Graz, 8036 Graz, Austria; 4Department of Physiology, Otto Loewi Center, Medical University Graz, 8036 Graz, Austria; 5Institute of Mechanical and Aerospace Engineering, University of California San Diego, La Jolla, CA 92093, USA; 6Physiology Division, Mohammed Bin Rashi University of Medicine and Health Sciences, Dubai 505055, United Arab Emirates

**Keywords:** lower body negative pressure, invasive hemodynamics, pressure-volume loops, preload reduction, Landrace pigs

## Abstract

Lower body negative pressure (LBNP) has been implemented as a tool to simulate systemic effects of hypovolemia, understand orthostatic challenges and study G load stress in humans. However, the exact hemodynamic mechanisms of graded LBNP followed by its abrupt release have not been characterized in detail, limiting its potential applications in humans. Here, we set out to investigate the immediate hemodynamic alterations occurring during LBNP in healthy Landrace pigs. Invasive cardiac monitoring via extensive pressure volume loop analysis was carried out during application of incremental LBNP up to life threatening levels from −15 to −45 mmHg as well as during its abrupt release. Three different sealing positions were evaluated. Incremental LBNP consistently induced a preload dependent depression of systemic hemodynamics according to the Frank-Starling mechanism. Overall, the pressure–volume loop progressively shifted leftwards and downwards with increasing LBNP intensity. The abrupt release of LBNP reverted the above-described hemodynamic changes to baseline values within only three respiratory cycles. These data provide quantitative translational insights into hemodynamic mechanisms of incremental and very high levels of LBNP, levels of seal and effect of abrupt release for future human applications, such as countermeasure development for long spaceflight.

## 1. Introduction

Lower body negative pressure (LBNP) is a well-established method to study orthostatic tolerance and conditions such as systemic hypovolemia or increased G-load stress and forces. Recently, nightly low-level, long-term LBNP (determined as 8 h) has been described as a potentially effective countermeasure against one of the most common microgravity-associated detrimental effects on human body, such as the “space-associated neuro-ocular syndrome [1]. However, the exact hemodynamic mechanisms of incremental LBNP to high (>30 mmHg) and commonly considered unsafe levels followed by its abrupt release have not been characterized in detail, limiting our understanding and thus its potential applications in humans.

During LBNP, blood is shifted from the heart towards the lower parts of the body in a controlled fashion [2,3,4,5,6]. The application of LBNP in humans usually consists of putting the individuals into a cylindrical, air-tight box, comparable to the one that was used during this study. The most commonly used sealing position in humans is at the iliac crest, even though other positions have also been tested. Negative pressures are then applied gradually, depending on the respective study protocol. It is also possible to vary the posture of the subjects into supine, standing or even head-tilted positions. [4,6] The amount of blood displaced increases, corresponding to the degree of negative pressure applied as well as to the location of the applied seal. It is known from previous studies that high levels of LBNP may cause central hypovolemia, declines in systemic blood pressure and cerebral perfusion and therefore loss of consciousness in humans [6,7,8,9,10]. However, all published studies so far related to LBNP have been carried out in humans and/or primates and are primarily limited by their non-invasive nature.

Similarly, hazardous levels of LBNP, causing mean aortic pressure to decrease below 30 mmHg, have never been applied in humans for obvious ethical reasons. Thus, an extensive characterization of cardiac hemodynamics during application of increasing amounts of LBNP up to the highest hemodynamically tolerable level, as well as at different sealing positions, is still needed to advance our limited knowledge.

Invasive pressure–volume assessment via a conductance catheter is considered the gold standard for directly measuring real-time, beat-to-beat LV function and mechanics [11]. We therefore set out to investigate the immediate hemodynamic alterations occurring during LBNP in acutely anesthetized healthy Landrace pigs, fully instrumented for LV and right ventricular (RV) assessment. Extensive LV pressure volume loop analysis, including end-systolic and end-diastolic relationship assessment as well as pharmacological beta-adrenergic stimulation to simulate cardiac stress, was carried out during application of incremental LBNP up to hazardous levels as well as during its abrupt release at three different sealing positions, moving from the upper abdomen down to the anterior iliac spine.

We hypothesized that (i) LBNP-induced hemodynamic changes would be most pronounced with a sealing position closest to the heart, while beta-adrenergic stimulation would counter its preload dependent effects on contractility and (ii) the hemodynamic instability induced by the highest tolerated LBNP pressure level would be fully reversible by abrupt LBNP release.

## 2. Materials and Methods

Experimental protocols were designed in accordance with the European Convention for the Protection of Vertebrate Animals used for Experimental and other Scientific Purposes and were approved by the Bioethics Committee of Vienna (Austrian Committee for Animal Trials, No. 2020-0.272.246).

### 2.1. Experimental Setup

The experimental setup has been described in detail before [12,13,14]. Briefly, all experiments were performed at the Section for Biomedical Research of the Medical University of Graz at identical time of day, location and room temperature. Nine Landrace pigs (68 ± 9 kg) were fasted overnight with free access to water. After the administration of 1 mg/kg of propofol via an intravenous line, the animals were intubated and anesthesia was continued as follows:0.25–0.5 mg/kg midazolam, 30–35 μg/kg fentanyl and 10–20 mg/kg ketamine intravenously, as well as 1.0–2.0 Vol% sevoflurane. Muscle relaxation was performed via a continuous infusion of pancuroniumbromid. Sheaths were introduced into both carotid arteries and internal jugular veins via an eight-centimeter median incision at the neck. At least one hour passed between the administration of the propofol bolus for induction of anesthesia and starting any hemodynamic measurements.

The animals were instrumented with a Swan-Ganz catheter (Edwards Lifesciences CCO connected to Vigilance I, Edwards Lifesciences, Irvine, CA, USA), an LV conductance catheter (5F, 12 electrodes, 7 mm spacing, MPVS Ultra, Millar Instruments, Houston, TX, USA) and a PTBV-balloon (Osypka VACS II^®^ 20 mm, Rheinfelden, Germany) in the distal descending aorta at the level of the diaphragm. Invasive instrumentation including pressure volume catheter in the left ventricle and the Swan Ganz catheter with its tip in the left pulmonary artery was confirmed during the intra-experimental X-ray (Figure 1).

Animals were placed in the LBNP chamber in supine position and were secured by fixing their upper legs to prevent them from being sucked into the chamber (Figure 2). LBNP was applied by means of a half cylinder plastic box and suction was applied using a standard vacuum cleaner that was connected to the distal end of the LBNP box via a 4 cm diameter tube. Air flow was regulated by an adjustable leak to pre-selected pressure steps and monitored by means of a manometer. A canvas skirt (similar to a kayak skirt) provided sealing around the animal’s body. After instrumentation, at least 30 min was allowed before baseline hemodynamic assessment, as obtaining a steady state was mandatory.

### 2.2. Experimental Protocol

The detailed protocol has been described in a recently published sub-study, focusing on the impact of LBNP on diastolic suction [14]. The LBNP sealing was performed at three different positions, as represented in Figure 2: I, 10 cm caudal of the xiphoid process; II, halfway between I and III; and III, at the height of the anterior iliac spine, which is the default LBNP sealing position in humans. Furthermore, hemodynamic assessment at position III was repeated during beta-adrenergic stimulation. Dobutamine was infused continuously after a single bolus and titrated to double the steady-state maximum LV dP/dt (position IIId), as previously published [12,13,14]. We clearly wanted to rule out any additional systemic effect of dobutamine between positions I to III and therefore chose to administer it only during position III, as this is also the position considered most comparable to the one used in protocols in humans.

Each protocol step (i.e., at any given sealing position) consisted of two phases of measurement: in phase I, LBNP was applied with the abovementioned increasing negative pressures (steps i–iii), while in Phase II hemodynamic assessment was performed during and after abrupt LBNP release. Whenever mean arterial pressure (MAP) decreased below 30 mmHg, suction was terminated and ambient pressure established immediately, as the resulting hemodynamic instability with high-risk of malignant, ventricular arrhythmias was considered “life-threatening” for the animal.

Baseline was recorded over three respiratory cycles and averaged, steady-state cardiovascular and hemodynamic measurements were continuously recorded (at each LBNP position). When the next negative pressure step was introduced at each position, a period of 10–15 min was waited to secure a steady state. If hemodynamics stayed consistently stable during measurements, we immediately moved to the next step. At the end of the study protocol, animals were euthanized by means of injection of 80 mL potassium chloride bolus.

To generate pressure–volume (PV) loops, afterload was increased briefly three times by inflating the intra-aortic balloon catheter in the descending aorta at each measurement step. Ventilation was halted during duration of during aortic occlusions, in order to minimize its impact on intrathoracic pressure and, therefore, biventricular interaction.

The detailed technical operating procedure of the conductance catheter and its calibration has been described before [15]. In short, the catheter is placed in the left ventricle via the aortic valve along the longitudinal axis. The two most distal and proximal electrodes generate an electric field. The eight electrodes in between divide the left heart into seven segments and each measures a conductance signal (Gi). Depending on the positioning of the catheter and heart size, five or more segments should lie in the ventricle. The positioning can be checked in the software by analyzing the PV-loops of each segment. The conductance of each segment corresponds to the volume and are summed to obtain the total ventricle volume. To convert the measured conductance into volume, the conductivity of the blood (σ) and the spacing between the electrodes (in our case: 7 mm) need to be considered. As the measured conductance is not limited to the blood in the cavity but also measures the conductance of the surrounding tissue such as the myocardium, this parallel conductance (Gp) needs to be determined. We used the method of hypertonic saline dilution [16]. A bolus of 4 mL hypertonic saline (NaCl 10%) was applied into the pulmonary artery to determine Gp. This corrected conductance is directly proportional to the actual volume but tends to underestimate. Therefore, a dimensionless constant α was introduced, which sets the conductance in relation to stroke volume (SV) measured by an independent source. We determined α by measuring Cardiac output (CO) continuously with the pulmonary artery flotation catheter.

### 2.3. Data Processing and Statistical Analyses

Data assessment and analysis were performed as previously described by our study group [13,14]. Briefly, PV-loops and time intervals were analyzed using CircLab Software (custom made by P. Steendijk [16]). The end-diastolic pressure-volume relationship (EDPVR) was derived from an exponential fit of end-diastolic pressure (EDP) and volume data points during aortic occlusion derived according to Ped = α × e β × Ved. The end-systolic pressure-volume relationship (ESPVR) was derived using a linear fit of end-systolic pressure and volume data points, characterized by the slope (end-systolic elastance [Ees]) and volume axis intercept (V0). In addition, to obtain single-point measures of PV-relationships, we calculated LV volumes at a LV-ESP pressure of 100 mmHg (LV VPes100). Isovolumic relaxation constant τ (ms) was calculated based on the method described by Raff and Glantz [17,18].

Figure 3 shows an example of representative PV-loops from one animal with all pressure levels and sealing positions, once again underlying the pronounced impact achieved by high amounts of LBNP at sealing position I and II compared to III and IIId.

All data are presented as mean ± SD. Multiple comparisons between measurement steps were compared by two-way ANOVA for repeated measurements. Post hoc testing was performed by Tukey’s test. Normality was tested using Shapiro–Wilks test. Non-normally distributed variables were examined either by Kruskal–Wallis or Friedman test. Significance was assumed when *p* < 0.05. For statistical calculations, we used Sigmastat (Version 4.0, Systat Software, Inc., San Jose, CA, USA) and SPSS (Version 23.0, IBM, Armonk, NY, USA).

## 3. Results

Eight of out of nine animals were included in the study and went through all pressure positions. One suffered from aortic dissection during inflation of the aortic balloon at the beginning of the protocol; therefore, it was completely excluded from the analysis. In another, aortic occlusion at step ii and iii was done at position I only, therefore the corresponding aortic occlusion-derived data were excluded from analysis completely, resulting in a number of specimen available for position I/step ii and iii of *n* = 7. Step IIId could not be performed in two animals due to hemodynamic compromise (data available for *n* = 6).

### 3.1. Phase I: Application of Increasing Lower Body Negative Pressure and Impact of Beta-Adrenergic Stimulation

LBNP was tolerated hemodynamically up to −45 mmHg at sealing positions II and III by all animals but by none at position I (cranial). No severe hemodynamic instability ensued at any sealing positioning with suction levels up to −30 mmHg. Heart rate (HR) remained unchanged (Figure 4A) at any pressure level of any sealing position, while increased at position III during dobutamine infusion. This might be related to the nature of the experiment in deep sedation. As expected, induced preload reduction gradually impacted on CO, which decreased (*p* < 0.05) significantly only at sealing position I and II, except for step—15 mmHg (Figure 4B). Similar trends were seen in respect to MAP, which markedly decreased at sealing position I (*p* < 0.05, Figure 4C). Mean central venous pressure (mCVP) decreased significantly from baseline in all positions when suction of at least −30 mmHg was applied, one step earlier in position I and II (Figure 4D). LV-EDP and LV-EDV (Figure 4E,F) significantly decreased from baseline at step iii (−45 mmHg) irrespective of sealing position, while, once again, significant decrease occurred already at a pressure of −15 mmHg at the cranial position. Overall, dobutamine infusion markedly impacted on LV volume and pressure changes, resulting in lower volumes and negative LV-EDP [14].

With regard to parameters related to LV systolic function, no statistically significant effects were observed in LV-end-systolic volumes (LV-ESV) (Figure 4G) or LV Ejection fraction (EF) (Figure 5A), potentially because of a parallel decrease in both LV-EDV and LV-ESV. Maximal LV pressure (LVPmax) decreased significantly at sealing positions I and II, while maximum rate of pressure-increase (LV-dP/dtmax) was reduced only at position I. (Figure 5B,C). Of note, baseline LV-dP/dtmax was doubled per definition during dobutamine infusion (IIId) and decreased over steps i-iii. Finally, LV-VPes100, a parameter derived by the ESPVR and less dependent on preload changes than dP/dtmax in reflecting LV contractility, did not show any change over the study protocol (Figure 5D).

As expected stroke work (SW) and the pressure volume area (PVA) decreased significantly at sealing positions I and II, while PVA only at position III (Figure 5E,F). Cardiac work efficiency (CWE) (Figure 5G) did not change significantly

### 3.2. Phase II: Release of Lower Body Negative Pressure

Figure 6 represents original pressure-volume recordings from Phase II, i.e., abrupt release of LBNP. LBNP release fully reverted the above-described left and downward shift of the pressure–volume loop (Phase I) back to the baseline values as quickly as within three respiratory cycles. HR remained substantially unaltered in Phase II (Figure 7A) as in phase I. CO and MAP returned to baseline levels throughout all measurement steps (Figure 7B,C), except from a slightly lower MAP compared to baseline at sealing position I. Mean CVP immediately recovered at each measurement step, as expected (Figure 7D).

LV-EDP, LV-EDV and LV-ESV returned from levels close to zero or even negative back to baseline (Figure 7E–G) (for more insights on negative LV-EDP refer to Berboth et al. [14], the sub-study focusing on diastolic suction). EF was once again unchanged (Figure 8A). LVPmax as well as LV-dP/dtmax recovered to baseline values (*p* < 0.05 in position I, Figure 8B,C). SW and PVA did not fully return to baseline (SW increased at seal position I, PVA at position II, Figure 8D,E). CWE remained unchanged (Figure 8F). The isovolumetric relaxation time constant τ did not change significantly throughout the protocol (Appendix A, online supplement). Dobutamine infusion did not significantly impact the observed phase II hemodynamic changes, except for the above-described difference in dP/dtmax.

## 4. Discussion

The exact hemodynamic mechanisms of graded LBNP followed by its abrupt release have not been characterized in detail, limiting its potential applications in humans. Here, we set out to investigate the immediate hemodynamic alterations occurring during LBNP in healthy Landrace pigs via extensive pressure–volume analysis. In summary, incremental LBNP causes preload-dependent hemodynamic alterations, up to life-threatening hypotension with hemodynamic instability. However, these changes are not associated with a detrimental effect on cardiac work efficiency, i.e., mechanical performance, and are steadily and fully reversible within three respiratory cycles after LBNP release. Beta-adrenergic stimulation does not significantly impact the preload dependent changes induced by LBNP on contractility. These data provide important quantitative insights into hemodynamic mechanisms of incremental LBNP at different sealing positions for future human applications, such as countermeasure development for long spaceflight.

Hemodynamic assessments of LBNP application in previous studies have been largely carried out non-invasively [6], commonly using pulse wave forms or Doppler probes, and therefore limiting their mechanistic accuracy. Furthermore, in human studies the occurrence of syncope due to reduced cerebral perfusion has been reported frequently [18,19]. However, the impact of hazardous low systemic pressures (MAP < 30 mmHg) on ventricular function and mechano-energetics remains largely unknown. As non-invasive measurements do not provide a comprehensive picture of ventricular hemodynamics during LBNP, there is a need to gain more in-depth insights via experimental studies to understand to what extent LBNP application is safe and reversible in humans [20]. For the first time, the present investigation employed high fidelity intracardiac micro-manometry and real-time measurements during graded LBNP up to life-threatening levels in anesthetized human-sized pigs. We evaluated the effect of different sealing locations and three negative pressures steps in order to take an in-depth view on hemodynamic changes during and immediately after LBNP. Looking at the results of this investigation, one has to bear in mind however, that this study was done under deep sedation of the specimens, and thus the impact of anesthesia on autonomic responses cannot be ruled out totally.

As expected, graded LBNP preload reduction induced a progressive decrease in systemic pressure, LV pressure and volumes, eventually inducing a significant CO decrease, according to the Frank-Starling mechanisms. Interestingly, we demonstrated a left- and downward shift of the PV-loop towards smaller volumes and pressures. This displacement was clearly happening along an unchanged ESPVR- and EDPVR-curve, as shown by an unchanged LV VPes100 (i.e., contractility) and, in a sub-study analysis, LV VPed10 (i.e., capacitance) [14]. LBNP therefore did not impact on LV contractility or LV compliance. LBNP induced acute preload reduction clearly mirrors the impact of a transient inferior vena cava occlusion, an established method to reduce preload in clinical and experimental settings, with the aim to assess ESPVR and EDPVR [21]. Application of dobutamine, titrated to double steady-state maximum LV dP/dt, increased contractility at baseline. However, its inotropic effect could not compensate the LBNP-induced loss of preload, with a similar drop in CO. In addition, it failed to induce a positive chronotropic effect, maybe biased by deep sedation. These results were in contrast with the initial hypothesis that beta-adrenergic stimulation could effectively counter contractility loss induced by preload reduction. Of note, we recently reported an increased LV suction phenomenon induced by dobutamine on top of LBNP, possibly related to enhanced diastolic suction [14]. The investigation of LBNP combined with dobutamine stress is novel. So far, human studies on LBNP have focused on the neuro-hormonal response to preload reduction, and in particular on systemic release of norepinephrine, vasopressin and renin-angiotensin at different LBNP levels [6]. Given a high reproducibility in the plasma response of these markers to LBNP-induced pre-syncope, LBNP was therefore suggested to be useful to predict subjects who are at higher risk of fainting.

The reduction in central blood volume clearly correlates with the position of the LBNP seal: A potentially life-threatening blood pressure decline (MAP < 30 mmHg) only occurred during LBNP at the highest level at the most cranial sealing position, i.e., −45 mmHg at position I. This is not surprising, as it has been reported that application of LBNP sealing at different locations influences hemodynamic responses. Moreover, including the splanchnic region in the LBNP suction area has also been shown to affect the hemodynamic responses [5,6,22]. Our data quantify the expected decrease in cardiac work and its almost complete recovery to baseline during phase II (release of LBNP). LBNP was tolerated up to −45 mmHg at sealing positions II and III, not at position I (cranial). Considering the rapid time course and experimental setup used, it seems reasonable to assume that the observed response patterns were solely driven by cardiac factors. However, in future studies, the roles of autonomic function changes and hormonal responses should be explored. For details regarding the changes seen at syncope in autonomic and hormonal parameters, please refer to Hinghofer-Szalkay et al. [23]. As we assume and conclude that as soon as venous blood returns to the heart, it is reestablished by ending short-term LBNP, we believe that there is not a relevant time correlation between LBNP duration and recovery from it.

With regard to LV mechano-energetics, external cardiac work, as measured by SW and PVA, progressively decreased with graded LBNP, while CWE, i.e., mechanical efficiency, did not change. LBNP application therefore reduces cardiac oxygen consumption, which is closely correlated to PVA [24], without compromising LV efficiency. This has important implications for future LBNP applications, hinting towards a lack of acute detrimental effects on cardiac performance. Furthermore, a sudden increase in preload after LNBP release re-established cardiac hemodynamics immediately. In studies with human participants, an almost immediate return of the hemodynamic parameters to baseline levels has been reported as well [6,23]. In conclusion, studies investigating the long-term effects of LBNP on cardiac performance are required to confirm these preliminary observations. The development of non-invasive, echocardiographic tools for assessment of cardiac work and efficiency [11], such as pressure–strain analysis, are therefore very suited to translate this concept into the human application.

LBNP is used as a substitute for simulating effects of central hypovolemia that occur during sudden blood loss or orthostatic challenge. Cardiac vagal withdrawal and reflex-mediated increase in sympathetic tone due to low-pressure stretch receptors can increase heart rate (HR) and catecholamine levels. Most investigators report LBNP-induced tachycardia [4,21,25,26,27,28,29,30]; occasionally, the opposite effect—bradycardia during LBNP—has been reported as well [31]. Bradycardia as participants develop syncope has previously been reported [32]. Overall, HR is a key parameter in human studies to assess autonomic response to LBNP application and its release. It was unanticipated, however, that the present examination did not show any LBNP related changes in HR. It appears reasonable, therefore, to assume that autonomic reflex mechanisms did not play a significant role in the hemodynamic changes observed in the current experimental study. This might be due to the anesthesia regimen, i.e., autonomic blockade due to the deep sedation of the animals, as well as to the lack of hydrostatic difference between their carotid baroreceptors and their cardio-pulmonary ones [33]. This might apply as well to neuroendocrine changes, i.e., release of endogenous catecholamines. Due to stable HR and the brevity of the LBNP stimuli applied (maximum 10–15 min), we assume neuro-hormonal effects did not influence observed hemodynamic changes due to lower body suction or its release.

### 4.1. Translational Outlook

LBNP has been primarily used to imitate systemic hypovolemia, orthostatic intolerance and the effects of G load force on humans. It has been adapted as well as exercise training (posture change, long-lasting bed rest, parabolic flight maneuvers and/or in spaceflight). Recently, nightly low-level, long-term LBNP has been described as a potentially effective countermeasure against one of the most common microgravity-associated detrimental effects on human body, such as “space-associated neuro-ocular syndrome” [1]. This present large animal study provides translational insights into the left ventricular and hemodynamic impact of incremental and very high levels of LBNP, levels of seal and effect of abrupt release for future human applications, such as countermeasure development for long spaceflight. We demonstrate no acute detrimental effect of LBNP on cardiac efficiency and fully reversibility of hemodynamic changes within 3 respiratory cycles. Nevertheless, acute hemodynamic instability and syncope during LBNP might cause serious short and long-term effects in an already impaired heart. Therefore, in populations that are particularly vulnerable to orthostatic tolerance and/or have common pre-existing cardiac impairment, such as people of older age, LBNP application should be carried out only under close surveillance.

### 4.2. Limitations

There are several limitations to our study. The relationship between LBNP and the volume pooled in the lower body is dependent on the size of the lower body, its own compliance and the individual vascular compliance itself. Pigs might possess higher venous volume capacity in their gluteus region compared to humans. However, previous experiments in baboons comparing the application of LBNP versus hemorrhage demonstrated that compensatory reserve responses are comparable between bi- and quadruped walking animals and are similar to those previously reported in humans during LBNP or hemorrhage [34].

In addition, the carotid baroreceptors of pigs reside approximately at the same arterial hydrostatic indifference level as their cardio-pulmonary ones; this could be mirrored in the missing reflex tachycardia. Due to ethical and legal requirements our study has been performed in anesthetized animals, which could potentially be another explanation for the lack of reflex tachycardia. It is known from investigations on non-sedated sheep, that the cardiac baroreflex response is the primary driver for compensation during LBNP [35], so this response might have been impaired in anesthetized pigs. Furthermore, we did not assess effects of the autonomic system and/or hormones on hemodynamic changes during short term LBNP [23]. It also needs to be pointed out, that we did not perform any neurologic monitoring of the animals during this study, neither were specimens for plasma hormone measurements taken.

Finally, as humans continue to explore deep space, and LBNP is routinely used in spaceflight as a countermeasure to improve blood flow to the lower body parts, there is a need to assess the potential impact of LBNP-related rheology on blood clotting. An ESA initiated Topical Team on Thrombo-embolism is currently examining how bed-rest confinement (a model for spaceflight induced deconditioning) and spaceflight as well as the application of countermeasures such as LBNP and artificial gravity can influence pro- and anti-coagulatory parameters in individuals of all ages [22,36,37].

## 5. Conclusions

In summary, incremental LBNP causes preload-dependent hemodynamic alterations, up to hazardous levels of systemic pressure accompanied by hemodynamic instability. However, these changes are not associated with a worse LV mechanical performance and are steadily reversible within three respiratory cycles after LBNP release. These data provide important mechanistic insights into the hemodynamic alterations of incremental LBNP at different sealing positions. Given the steadily increasing fields of LBNP application in human research, such as spaceflight research, future translation of these data is granted.

## Figures and Tables

**Figure 1 jcm-11-05858-f001:**
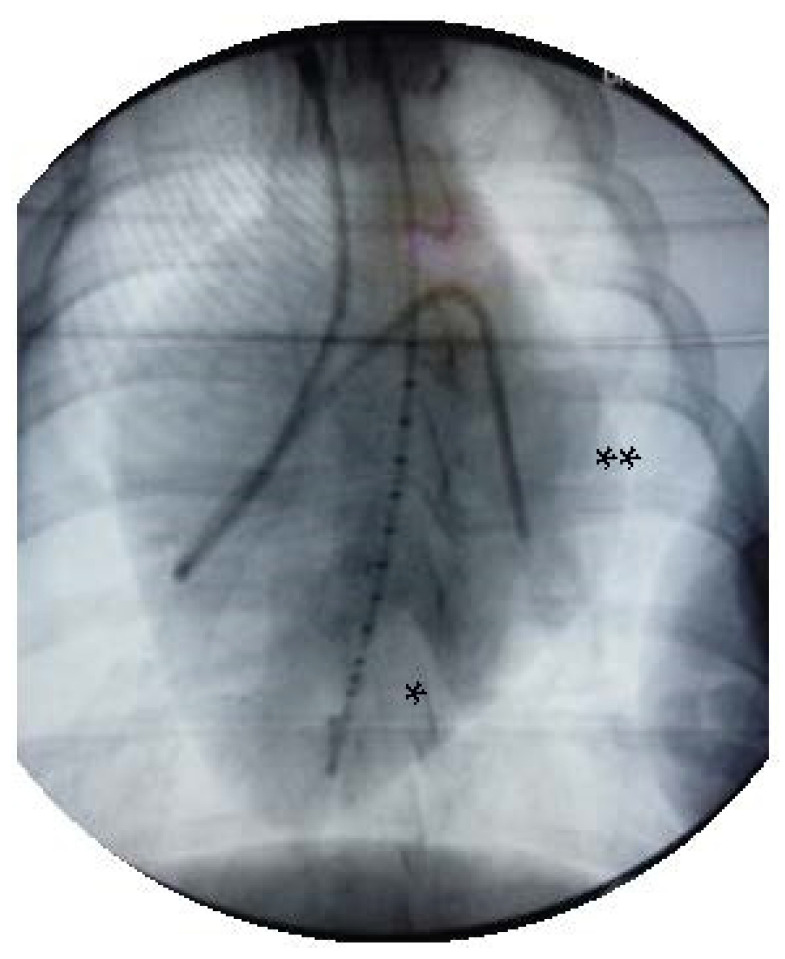
X-ray after invasive instrumentation including pressure volume catheter in the left ventricle (*) and the Swan Ganz catheter with its tip in the left pulmonary artery (**).

**Figure 2 jcm-11-05858-f002:**
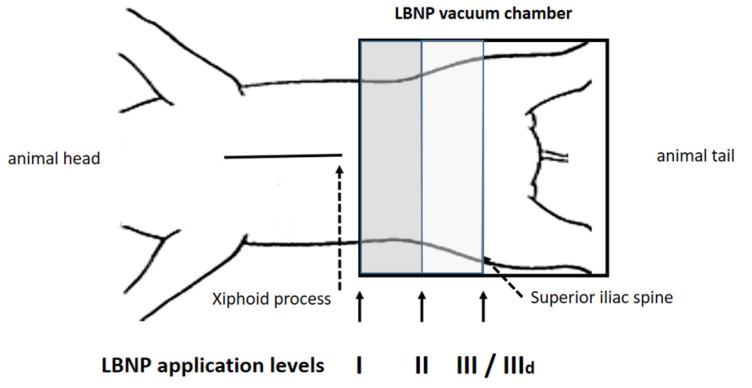
Position I was 10 cm below the xiphoid process, position III at the anterior iliac spine, position II between I and III. After reaching a steady state baseline recordings were performed, followed by measurements at −15, −30 and −45 mmHg (steps i–iii). At position III LBNP was repeated under dobutamine infusion (IIId).

**Figure 3 jcm-11-05858-f003:**
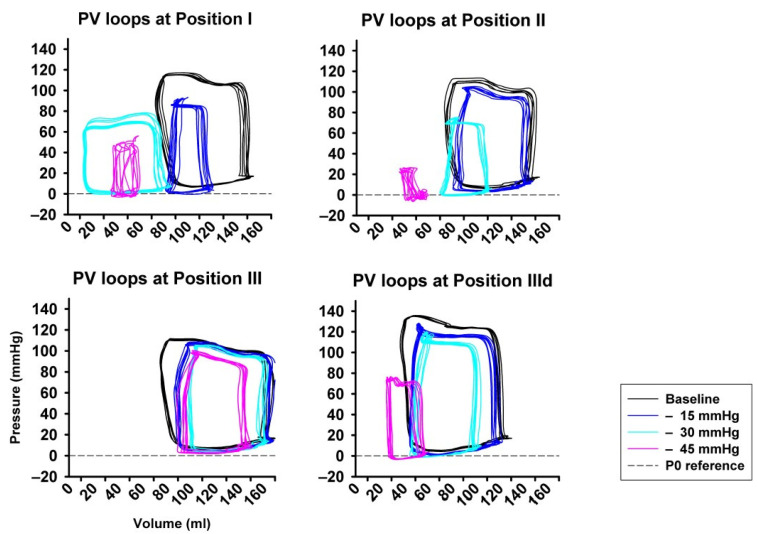
Examples from original pressure–volume loops (x-axis: volume in ml, y-axis: pressure in mmHg) for each sealing position (I–III) and pressure step (i−iii), visually showing the pronounced impact of LBNP-derived preload reduction. A left- and down-ward shift of the loops is more pronounced at sealing position I compared to position II and III.

**Figure 4 jcm-11-05858-f004:**
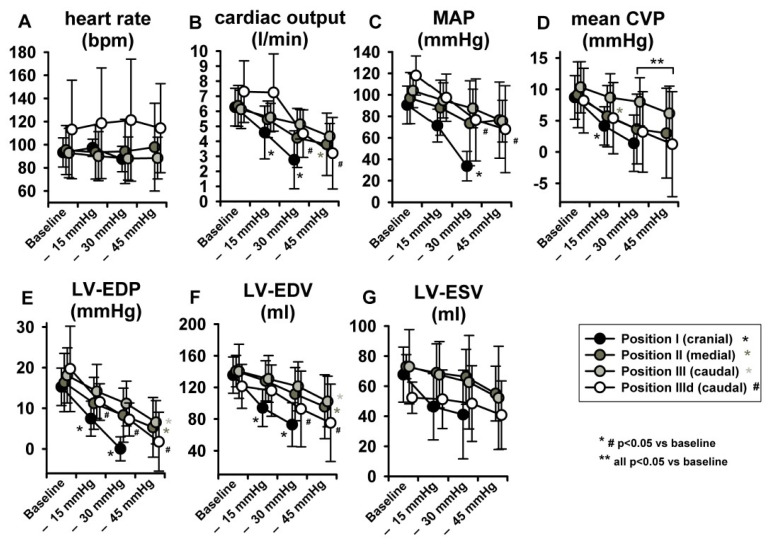
LBNP did not elevate heart rate (HR). Significant LBNP-induced decrease was seen in cardiac output (CO), mean arterial pressure (MAP), mean central venous pressure (mCVP), left ventricular end-diastolic pressure (LV-EDP) and left ventricular end-diastolic volume (LV-EDV). Left ventricular end-systolic volume (LV-ESV) did not change significantly.

**Figure 5 jcm-11-05858-f005:**
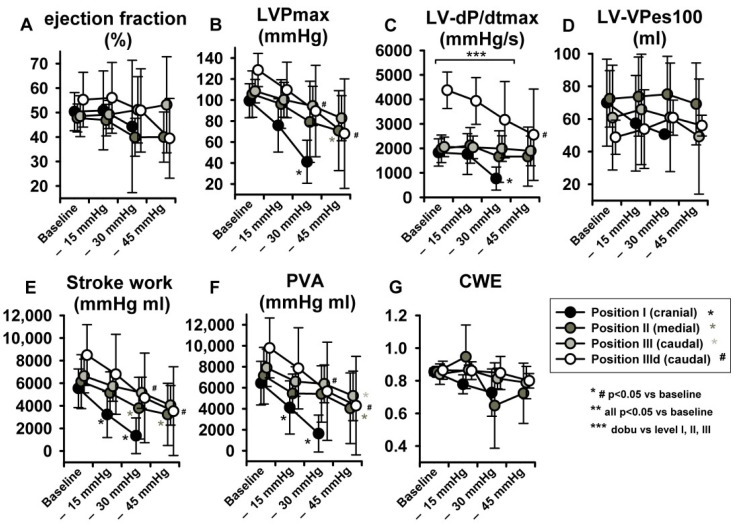
LBNP negatively impacted maximal left ventricular pressure (LVPmax) (**B**), maximum rate of pressure increase (LV-dP/dtmax) (**C**), stroke work (SW) (**E**) and pressure volume area (PVA) (**F**). No significant effect ensued in ejection fraction (EF) (**A**), left ventricular-end-systolic volume at 100 mmHg (LV-VPes100) (**D**) and cardiac work efficiency (CWE) (**G**). Data are presented as average with error bars indicating SD. Asterisk (for position I-III) and # (for dobutamine, position IIId) mean *p* < 0.05 compared to baseline.

**Figure 6 jcm-11-05858-f006:**
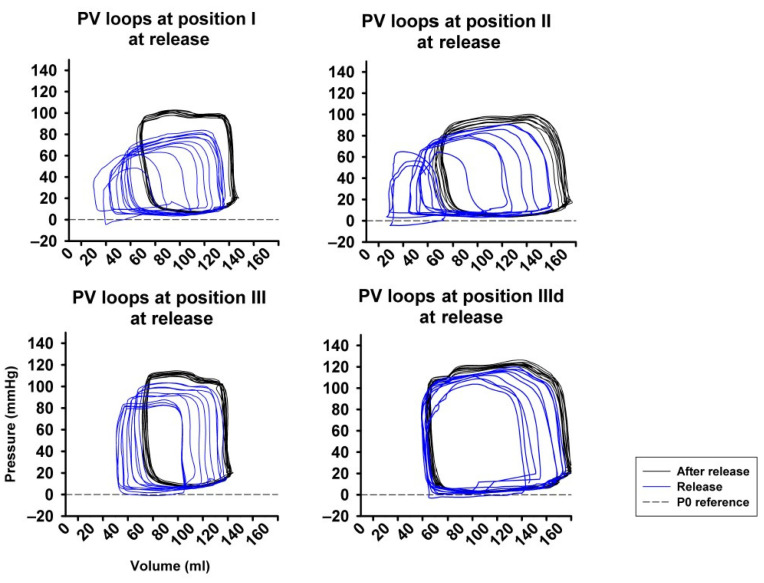
Full return of cardiac volume load after abrupt release of LBNP demonstrated by the right- and upward shift of the pressure-volume loop (PV loop), more pronounced at position I compared to position II and III. P0 reference marks the reference line at zero pressure.

**Figure 7 jcm-11-05858-f007:**
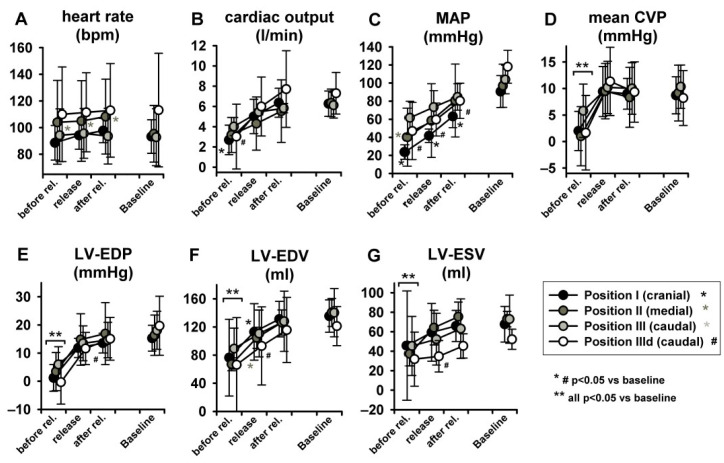
Full return of cardiac volume load after three respiratory cycles after LBNP release (MAP = mean arterial pressure, mean CVP = mean central venous pressure, LV-EDP = left ventricular-end-diastolic pressure, LV-EDV = left ventricular end-diastolic volume, LV-ESV = left ventricular end-systolic volume. Heart rate did not change (as statistic spike it did in position 2).

**Figure 8 jcm-11-05858-f008:**
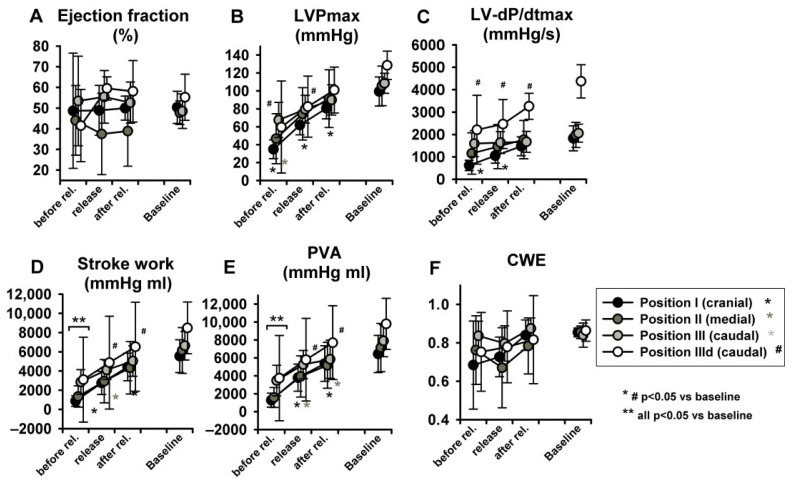
Hemodynamic changes after abrupt release of LBNP: The LV ejection fraction (**A**), maximum of left ventricular pressure (LVPmax) (**B**), maximum rate of pressure increase (LV-dP/dtmax) (**C**) and PVA (**E**) increased to baseline values. Stroke work (**D**) and pressure volume area (PVA) did not. Cardiac work efficiency (CWE) (**F**) was unchanged. Asterisk and # indicate *p* < 0.05 compared to baseline, ** indicates that at this level all positions are *p* < 0.05 to baseline.

## Data Availability

The data that support the findings of this study are available from the corresponding author upon reasonable request.

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
