# Peer review of "Impact of Increasing Lower Body Negative Pressure and Its Abrupt Release on Left Ventricular Hemodynamics in Anesthetized Pigs"

_jcm, 2022, doi:10.3390/jcm11195858_

Round 1
Reviewer 1 Report
The authors present the results of an in vivo study of the effects of graded LBNP and sudden release at various sealing positions on pigs. This study seems well designed, and it is unique in its use of invasive measurement techniques to gather previously unattainable data. Data interpretation seems well thought out and thorough. The results regarding the effects of LBNP levels and sealing positions on numerous cardiac parameters are very interesting. Overall, I feel this is a strong manuscript and I recommend publication after minor revision.
Suggestions/Comments:
- Lines 120-131: Please include information about the duration of this protocol. You mention that 30 minutes are allowed between instrumentation and baseline (I assume at each sealing position), but there is no mention of how long it takes to step through the LBNP applied pressures. In lines 340-341, you do mention ‘brevity of the LBNP stimuli’, but it would be useful to the reader to provide at least an approximate duration for each position. I believe it is important to know this, because the recovery after LBNP release that you note in your results may depend on the brevity of the LBNP application.
- Lines 163-168: The description of animals included in the study is somewhat unclear. You state that 8/9 animals were included and went through all pressure positions. You mention one pig suffered aortic dissection…was this #9, or did this occur part way through measurements in one of the other pigs? You state that 8/9 went through all pressure positions, but then say that Step IIId could not be performed in two animals. Please clarify. It would help if you would conclude this paragraph with something like: Thus, the results for Positions I, II and III are for n=8, and the results for Position IIId are for n=6.
- Lines 191 – 202: Swap Fig. 4 and Fig. 5…it does not make sense to talk about Fig. 3, then Fig. 5, then Fig. 4.
- Line 195: The word “increase” in this line is confusing. The plot shows a decrease in the “maximum rate of pressure increase” at Position I as LBNP becomes more negative. To clarify, perhaps change the wording of this sentence to ‘…while maximum rate of pressure rise (LV-dP/dtmax) only decreases at position I.’
- Line 198: Change ‘dP/dptmax’ to ‘dP/dtmax’
- Line 217: Change ‘Fig. 4’ to ‘Fig. 5g’
- Line 268: Change ‘known’ to ‘unknown’
- Lines 280 – 283: this is a long, complicated, hard to read sentence. I suggest putting parentheses around ‘(i.e., contractility)’ and ‘(i.e., capacitance)’ to help clarify.
- English in the manuscript is reasonable but could use improvement in places. For example: remove the word ‘a’ in line 200; replace ‘pronounce’ with ‘pronounced’ in line 201; remove the word ‘on’ in lines 248, 259, and 283; change ‘assessment’ to ‘assessments’, and change ‘have been’ to ‘were’ in line 263; change ‘which’ to ‘what’ in line 271. Many more small examples of this throughout the manuscript.
Author Response
Dear Reviewer 1,
Dear Sir or Madame,
Thanks for your kind remarks and suggestions for improvement of our manuscript on LBNP.
We responded to our best knowledge to your pending questions, the paper was corrected and re-read by various co-authors. The Line No. used in describing the response to your remarks refers to the manuscript version with visible "mark ups", so it would be easier to follow the changes made.
We hope that we were able to improve the manuscript and would be delighted if it was considered for publication by your esteemed journal.
Thanks a lot in advance,
Sincerly,
Birgit Zirngast
- Lines 120-131: Please include information about the duration of this protocol. You mention that 30 minutes are allowed between instrumentation and baseline (I assume at each sealing position), but there is no mention of how long it takes to step through the LBNP applied pressures. In lines 340-341, you do mention ‘brevity of the LBNP stimuli’, but it would be useful to the reader to provide at least an approximate duration for each position. I believe it is important to know this, because the recovery after LBNP release that you note in your results may depend on the brevity of the LBNP application.
Line 155 was added: “When the next negative pressure step was introduced at each position a period of 10-15 minutes was awaited to secure steady state. If hemodynamics stayed consistently stable during measurements, we immediately moved on the the next step.”
- Lines 163-168: The description of animals included in the study is somewhat unclear. You state that 8/9 animals were included and went through all pressure positions. You mention one pig suffered aortic dissection…was this #9, or did this occur part way through measurements in one of the other pigs? You state that 8/9 went through all pressure positions, but then say that Step IIId could not be performed in two animals. Please clarify. It would help if you would conclude this paragraph with something like: Thus, the results for Positions I, II and III are for n=8, and the results for Position IIId are for n=6.
Line 226 changed to: “One suffered from aortic dissection during inflation of the aortic balloon at the beginning of the protocol, therefor it was completely excluded from the analysis. In another, aortic occlusion at step ii and iii was done at position I only, therefore the corresponding aortic occlusion-derived data were excluded from analysis completely, resulting in a number of specimen available for position I/step ii and iii of n = 7. Step IIId could not be performed in two animals due to hemodynamic compromise (data available for n = 6).”
- Lines 191 – 202: Swap Fig. 4 and Fig. 5…it does not make sense to talk about Fig. 3, then Fig. 5, then Fig. 4.
Line 207: Figure 3 was changed to “Figure 4” and vice versa, corresponding references in text were changed; Figure was put into different place in the text to optimize flow of reading.
- Line 195: The word “increase” in this line is confusing. The plot shows a decrease in the “maximum rate of pressure increase” at Position I as LBNP becomes more negative. To clarify, perhaps change the wording of this sentence to ‘…while maximum rate of pressure rise (LV-dP/dtmax) only decreases at position I.’
Line 262 : “maximum rate of pressure increase” was changed to “maximum rate of pressure-increase”.. meaning “maximum change in increase of pressure” got reduced
- Line 198: Change ‘dP/dptmax’ to ‘dP/dtmax’
Line 265: was adapted
- Line 217: Change ‘Fig. 4’ to ‘Fig. 5g’
Line 207: was adapted; corresponding references in text changed
- Line 268: Change ‘known’ to ‘unknown’
Line 336: was adapted
- Lines 280 – 283: this is a long, complicated, hard to read sentence. I suggest putting parentheses around ‘(i.e., contractility)’ and ‘(i.e., capacitance)’ to help clarify.
Line 351: was adapted
- English in the manuscript is reasonable but could use improvement in places. For example: remove the word ‘a’ in line 200; replace ‘pronounce’ with ‘pronounced’ in line 201; remove the word ‘on’ in lines 248, 259, and 283; change ‘assessment’ to ‘assessments’, and change ‘have been’ to ‘were’ in line 263; change ‘which’ to ‘what’ in line 271. Many more small examples of this throughout the manuscript.
- Manuscript was re-read by various co-authors and re-checked for improvement of language.
Reviewer 2 Report
I read with great interest the work of Zirngast et al. They performed a really well organized animal study and therefore they could provide some interesting information regarding the haemodynamic impact of LBNP that seem to advance our knowledge. I have a few minor comments for the authors.
Introduction
Line 42: Please provide a brief description of the technical method used for LBNP in humans.
Line 44: Please provide the method used for LVEDV, SV and CO assessment in the articles that you refer to.
Methods
Please provide how long after the administration of iv propofol you performed your measurements, as this drugs are known to depress cardiac contractility.
Please comment why you decided to administer dobutamine only in the position III.
Results
Line 175: Please clarify if CO significantly decreased (compared to baseline) for all levels of pressure applied at each position. According to the graphs provided it seems tthat there was no significant decrease for -15mmHg at positions II, III.
Discussion
Paragraph 1: Please comment that these results should be interpreted under the prisma of sedation. As no HR increase was noted with the parallel decrease of preload, this is far from the physiologic response and may bias the results.
Line 286: Please comment again that although beta adrenergic stimulation increased contractility it failed to induce a positive chrotropic response.
Paragraph 4: Do the authors believe (or is there any relevant literature) that the time of LBNP application could influence the time needed to return to baseline (ie prolonged LBNP> prolonged time to recover).
Limitations: Please acknowledge again as limitations that there was no neurologic monitoring of the pigs as well as that plasma hormones were not measured
Author Response
Dear Reviewer 2,
Dear Sir or Madame,
Thanks for your kind remarks and suggestions for improvement of our manuscript on LBNP.
We responded to our best knowledge to your pending questions, the paper was corrected and re-read by various co-authors. The line no. describing the response to your remarks in our reply refers to the manuscript version with visible “mark ups” so it would be easier to follow the changes made.
We hope that we were able to improve the manuscript and would be delighted if it was considered for publication by your esteemed journal.
Thanks a lot,
Sincerely,
Birgit Zirngast
Introduction
Line 42: Please provide a brief description of the technical method used for LBNP in humans.
Line 44 was added: “The application of LBNP in humans usually consists of putting the individuals into a cy-lindrical, air-tight box, comparable to the one that was used during this study (Fig. 2.). The most common used sealing position in humans is at the iliac crest, even though, other po-sitions haven been tested out too. Negative pressures are then applied gradually, depend-ing on the respective study protocol. It is also possible to vary the posture of the subjects into supine, standing or even head-tilded positions.”
Line 44: Please provide the method used for LVEDV, SV and CO assessment in the articles that you refer to.
Line 50 :” LBNP induced preload reduction leads to a decrease in Left ventricular end-diastolic volume (LV-EDV), stroke volume (SV) and therefore cardiac output (CO), in accordance to the Frank-Starling mechanism.” was deleted as refers to present manuscript and does not appear reasonable in this part of paper.
we added instead in Line 173: “The detailed technical operating procedure of the conductance catheter and its calibration has been described before [15]. In short, the catheter is placed in the left ventricle via the aortic valve along the longitudinal axis. The two most distal and proximal elec-trodes generate an electric field. The eight electrodes in between divide the left heart into seven segments and each measures a conductance signal (Gi). Depending on the positioning of the catheter and heart size five or more segments should lie in the ventricle. The positioning can be checked in the software by analyzing the PV-loops of each segment. The conductance of each segment corresponds to the volume and are summed to obtain the total ventricle volume. To convert the measured conductance into volume, the conductivity of the blood (σ) and the spacing between the electrodes (in our case: 7mm) need to be considered. As the measured conductance is not limited to the blood in the cavity but also measures conductance of surrounding tissue like myocardium, this parallel conductance (Gp) needs to be determined. We used the method of hypertonic saline dilution [16]. A bolus of 4ml hypertonic saline (NaCl 10%) was applied into the pulmonary artery to deter-mine Gp. This corrected conductance is directly proportional to the actual volume but tends to underestimate. Therefore, a dimensionless constant α was introduced, which sets the conductance in relation to stroke volume (SV) measured by an independent source. We determined α by measuring Cardiac output (CO) continuously with the pulmonary artery flotation catheter.”
Methods
Please provide how long after the administration of iv propofol you performed your measurements, as this drugs are known to depress cardiac contractility.
Line 95: was added “At least one hour passed between the administration of the propofol bolus for induction of anesthesia and starting any hemodynamic measurements.”
Please comment why you decided to administer dobutamine only in the position III.
Line 140: was added “We clearly wanted to rule out any additional systemic effect of dobutamine between posi-tions I to III and therefore chose to administer it only during position III, as this is also the position considered most comparable to the one used in protocols in humans.”
Results
Line 175: Please clarify if CO significantly decreased (compared to baseline) for all levels of pressure applied at each position. According to the graphs provided it seems tthat there was no significant decrease for -15mmHg at positions II, III.
Line 243: was corrected to “except for step – 15 mmHg”
Discussion
Paragraph 1: Please comment that these results should be interpreted under the prisma of sedation. As no HR increase was noted with the parallel decrease of preload, this is far from the physiologic response and may bias the results.
Line 344: “Looking at the results of this investigation, one has to bear in mind however, that this study was done under deep sedation of the specimens and thus an impact of anesthesia on autonomic responses cannot be ruled out totally.”
Line 286: Please comment again that although beta adrenergic stimulation increased contractility it failed to induce a positive chrotropic response.
Line 360: was added “In addition, it failed to induce a positive chronotropic effect, maybe biased by deep sedation.”
Paragraph 4: Do the authors believe (or is there any relevant literature) that the time of LBNP application could influence the time needed to return to baseline (ie prolonged LBNP> prolonged time to recover).
Line 384: was added “As we assume and conclude, that as soon as venous blood return to the heart is reestab-lished by ending short-term LBNP, we believe, that there is not a relevant time correlation between LBNP duration and recovery from it.”
Limitations: Please acknowledge again as limitations that there was no neurologic monitoring of the pigs as well as that plasma hormones were not measured
Line 353: was added “It also needs to be pointed out, that we did not perform any neurologic monitoring of the animals during this study, neither were specimens for plasma hormone measurements taken.”
The manuscript was re-read by various co-authors and re-checked for improvement of language as well.